# Recent Advances in Polymer-Based Nanomaterials for Non-Invasive Photothermal Therapy of Arthritis

**DOI:** 10.3390/pharmaceutics15030735

**Published:** 2023-02-22

**Authors:** Muktika Tekade, Prashant Pingale, Rachna Gupta, Bhakti Pawar, Rakesh Kumar Tekade, Mukesh Chandra Sharma

**Affiliations:** 1School of Pharmacy, Devi Ahilya Vishwavidyalaya, Takshila Campus, Khandwa Road, Indore 452001, Madhya Pradesh, India; 2Department of Pharmaceutics, Sir Dr. M.S. Gosavi College of Pharmaceutical Education and Research, Nashik 422005, Maharashtra, India; 3National Institute of Pharmaceutical Education and Research (NIPER), Ahmedabad, Palaj, Opp. Air Force Station, Gandhinagar 382355, Gujarat, India

**Keywords:** polymer, nanomaterials, photothermal, arthritis, near infra-red, inflammation, drugs

## Abstract

To date, nanomaterials have been widely used for the treatment and diagnosis of rheumatoid arthritis. Amongst various nanomaterials, polymer-based nanomaterials are becoming increasingly popular in nanomedicine due to their functionalised fabrication and easy synthesis, making them biocompatible, cost-effective, biodegradable, and efficient nanocarriers for the delivery of drugs to a specific target cell. They act as photothermal reagents with high absorption in the near-infrared region that can transform near-infrared light into localised heat with fewer side effects, provide easier integration with existing therapies, and offer increased effectiveness. They have been combined with photothermal therapy to understand the chemical and physical activities behind the stimuli-responsiveness of polymer nanomaterials. In this review article, we provide detailed information regarding the recent advances in polymer nanomaterials for the non-invasive photothermal treatment of arthritis. The synergistic effect of polymer nanomaterials and photothermal therapy has enhanced the treatment and diagnosis of arthritis and reduced the side effects of drugs in the joint cavity. In addition, further novel challenges and future perspectives must be resolved to advance polymer nanomaterials for the photothermal therapy of arthritis.

## 1. Introduction

Rheumatoid arthritis (RA) is a chronic autoimmune condition that is correlated with a high death rate and affects nearly 1% of the population worldwide. It is distinguished by the infiltration of inflammatory cells, synovial hyperplasia, pannus development, persistent cartilage, and bone deterioration [1]. The inflammatory response, predominantly in RA, deals with immune cells such as macrophages, neutrophils, and T cells, which secrete a large number of pro-inflammatory cytokines and also generate reactive oxygen species (ROS) that cause structural damage to cartilage and weaken the functional ability in the arthritic area (Figure 1). Due to the breakdown of bone and cartilage, joint swelling, and discomfort, this chronic autoimmune inflammatory disease limits movement and lowers quality of life. The relevance of RA diagnosis and therapy is critical because obtaining an accurate, early diagnosis of RA can be challenging, while the lack thereof can be detrimental to RA treatment. Many disease-modifying anti-rheumatic drugs (DMARDs) such as methotrexate (MTX), dexamethasone (Dex), tocilizumab (TCZ), and others have been used to reduce joint inflammation and deformity. Even so, these drugs have not been proven helpful for patients due to their adverse effects following long-term use [2].

The quality of life for many patients has increased thanks to recent developments and novel treatment modalities that have significantly weakened disease progression [3]. The advancement of nanotechnology opens fresh perspectives for the study and treatment of RA. Using nanomaterials (NMs) as nanocarriers for drugs has enhanced their bioavailability and bioactivities and upgraded the selective targeting of inflamed joints. In addition, NMs’ small size and considerable surface activity have enabled the controlled release of drugs that reduces their dose-dependent side effects. Combining diagnosis and treatment is the most effective strategy for managing the condition of RA patients. Recent research has demonstrated improved arthritis treatment via the employment of the combined effect of nanomaterial and photothermal therapy (PTT). Among various NMs, polymer-based NMs act as vital photothermal reagents with high absorption in the near-infrared (NIR) region that can transform near-infrared light into localised heat, thus offering fewer side effects, easier integration with existing therapies, and increased effectiveness [4]. NMs with PTT can easily penetrate joint tissues without causing damage and facilitate imaging for an arthritis diagnosis. Recently, manifold works have been carried out that have revealed the efficiency of the combinatory effect of polymer-based NMs and PTT with respect to the treatment and diagnosis of RA [5]. This review article will discuss the advancement of polymer-based NMs for non-invasive PTT in arthritis.

Healthy bone is made up of two neighbouring bones that are covered in cartilage. The articular cavity is the area between the ends that carries synovial fluid and is delineated by the synovial membrane on both sides. The synovial membrane is a thin layer of cells made up of two types of synovial cells: type A, which resemble macrophages, and type B, which resemble fibroblasts. Due to the porous structure of the synovial membrane, nutrients in the serum can diffuse cartilage while also producing synovial fluid. The development of rheumatoid arthritis and immunological reactions are complicated and involve interaction between elements of innate and adaptive immune systems. Numerous inflammatory cells are present in the synovial membrane, where they interact to promote joint degeneration [6]. The existence of dendritic cells, a significant class of antigen-presenting cells (APC) that produces several cytokines, suggests the significance of the adaptive immune pathway with respect to RA. The mechanism of T-cell stimulation includes dendritic cells, which also provide antigens to T cells already existing in the synovium [7]. There are two signals needed for T-cell activation: the presentation of an antigen to the T-cell receptor is the preliminary signal, while the CD80/86 surface protein present on the dendritic cell must connect with the CD28 protein of the T cell for the second signal, which is known as the costimulatory signal [8]. The activation of T-cells cause macrophages and dendritic cells to secrete different TGF-β and interleukins (IL-1,6,17, 21, and 23) and reduces the generation of regulatory T-cells, hence leading to inflammation in the synovial fluid. In addition to this, the activation of T-cells also causes the release of IL-17, which stimulates fibroblasts and macrophages such as synovial cells for the activation of IL-26, in turn leading to the further production of IL-6, TNF-α, and IL-1β, thereby causing inflammation [9].

Another mechanism involved in RA is an antigen-nonspecific pathway that requires cell-to-cell interaction between fibroblasts and macrophages with activated T-cells [10]. Adaptive humoral immunity is also a significant component of the aetiology of RA. Multiple potential strategies may be used to mediate the role of B cells in autoimmune disorders. Additionally, B cells can produce anti- and pro-inflammatory cytokines, which may be involved in the generation of RA [11]. An overview of the abovementioned arthritis pathogenesis is provided in Figure 1. 

## 2. Fundamentals of Photothermal Therapy

PTT uses the near infra-red wavelength (NIR; 780 nm to 2500 nm) of the electromagnetic spectrum to treat diseases such as cancers and arthritis. Upon exposure to NIR, the tissue temperature increases due to the conversion of light energy to heat, thereby killing cancerous cells or tissue. Three photophysical steps are involved in the conversion of NIR energy into heat. In the first and second steps, upon absorbing light, NMs convert into hot lattices, resulting in charge separation particle oscillation. Once light has been absorbed and transformed into heat, electron–photon relaxation occurs. The final step is cooling, during which heat is emitted from metal NMs into the surrounding environment. (Figure 2). Thus, such energy is utilised in arthritis treatment that causes cell demolition and apoptosis.

Owing to the small size of NMs, the restriction of electronic motion differs from that of bulk material. When light energy is incident on the NMs, the electrons on the NMs’ surface become excited, and an electron cloud forms over this surface. Here, the free electrons transfer from the valence band to the conduction band, which leads to the enhancement of the resonance peak. NMs, including graphene, carbon nanotubes, and gold nanoparticles (NPs), are the subject of preclinical investigation as possible agents for the photothermal treatment of cancer, particularly tumours. However, there are still obstacles to be overcome to improve light conversion efficiency and decrease damage to healthy cells and tissues. A new class of drugs called biological response modifiers targets inflammatory mediators in the treatment of RA. At optical frequencies, metal NPs effectively scatter and absorb visible light. Recent advancements in the synthesis, conjugation, and fabrication of NPs have prompted a greater interest in employing plasmon-resonant NPs. Additionally, their optical and photonic usage for biological purposes has been developed (Tank ND, 2017). To control biological autoimmune systems and treat diseases, a novel therapeutic strategy has recently arisen due to its potential to minimize side effects. 

## 3. Various Polymer-Based Nanomaterials for Non-Invasive Photothermal Therapy of Arthritis

### 3.1. Polymer-Based Nanomaterials 

In recent decades, conjugated, polymer-based nanomaterials have been used in photothermal therapy [12]. Conjugated polymers possess recurring chemical units, delocalised electron systems, and conjugated bonding. These polymer-based nanomaterials (NMs) are biocompatible, cost-effective, biodegradable, possess high photothermal conversion efficiency, and are efficient nanocarriers for the delivery of drugs to a target cell. Furthermore, they can easily be surface-modified and functionalised with numerous functional groups and can encapsulate therapeutic agents. Examples of such polymers include polypyrrole, polyaniline, and polythiophene, which have been extensively used in biomedical applications. 

So far, numerous switches have been used to control a drug’s release from the formulation. Amongst them all, the use of a light-triggered sequential release system stands out from the rest of the stimuli. The combination of the use of conjugated, polymer-based nanomaterial with laser therapy can achieve controlled drug release. The release of an anti-rheumatic drug incorporated into the polymeric material upon laser irradiation is depicted in Figure 3. The combined effect of PTT and polymer NMs have improved the treatment and diagnosis of RA and reduced the side effects of drugs in the joint cavity. Various reports exhibiting the efficacy of polymeric-based NMs for RA treatment are available. 

For instance, polyethylene glycol (PEG), primarily, is used for the surface modification of NMs. In one study, Quindeel and co-workers prepared a polycaprolactone–polyethylene glycol–polycaprolactone (PCL-PEG-PCL) triblock copolymer via a ring-opening copolymerization reaction and utilised such polymer NMs as nano vehicles for the delivery of the drug MTX. Their results revealed the excellent efficacy of NMs with respect to an RA mouse model compared to those models of free MTX. These fabricated NMs decrease the expression of the inflammatory factors and recover the behavioural response capability of mice [13]. 

Another study by Chen and colleagues [14] demonstrated the efficacy of photothermally triggered nitric oxide (NO) nanogenerators via combining photothermal agents and NO within polymeric NMs. NIR light within 650–900 nm is an excellent trigger-responsive system for the sustained release of NO. The authors synthesised NO-Hb@siRNA@PLGA-PEG (NHsPP), wherein hemoglobin (Hb) acts as a carrier for NO that absorbs NIR at 650 nm, and siRNA was also used in combination for the inhibition of the inflammatory response through the Notch signalling pathway. Their results exhibited controlled NO release in RAW 265.7 cells and collagen-induced arthritis (CIA) mice. Moreover, by combining NO, siRNA, and PTT, an excellent macrophage inflammatory response was shown that could be used for other inflammatory diseases [14]. A few of the other studies are discussed in Table 1.

Another study by Lee et al. [15] revealed the effect of MTX-loaded PLGA NMs on RA treatment by conjugating them with NIR irradiation. Their nanocomposite showed an excellent therapeutic outcome compared with MTX alone in CIA mice and complete Freund’s adjuvant-induced arthritic mice (Figure 4). Their results also improved the controlled release of MTX and its bioavailability to the specific arthritic site using the developed functional polymer-NIR bioconjugate nanocomposite [15]. 

### 3.2. Polymer-NIR780 Bioconjugate

IR780, a lipophilic, cationic, heptamethine dye with an emission wavelength of 780 nm, generates heat upon its irradiation with NIR light. NIR780 is a NIR bioactive dye that is usually conjugated with a polymer for photothermal therapy [16]. PTT can target specific arthritic sites and produce heat due to interaction of such NIR active dye with light. The combinatory effect of PTT and polymer NMs has enhanced absorption in the NIR region (700–950 nm) [17]. Using a polymer–NIR 780 bioconjugate, light irradiation triggers the heat generation response of the material that serves as a mediator to activate structural polymer modifications, and this process, in turn, offers novel material functionalities such as the switching of drug delivery on/off or signals providing the basis for imaging. Together with PTT, such polymer bioconjugate materials can be used for the treatment and diagnosis of arthritis issues. 

For example, in one report, researchers developed a nanogold core multifunctional dendrimer nanocomposite that was conjugated with the drug MTX, wherein the latter acts as a targeting ligand for receptors in arthritic tissues. In addition, NIR-780 was also loaded with this nanocomposite to provide PTT via light irradiation. Their results showed the sustained release of MTX and the enhanced anti-inflammatory efficacy of the nanocomposite via its combination with PTT towards mouse macrophage RAW 265.7 cells and CIA mice model. Hence, their multifunctional, targeted nanocomposite proved to be a potent therapeutic strategy for improved RA therapy [16]. 

### 3.3. Polymer-Based Gold Nanomaterials 

Gold nanoparticles number among the FDA-approved metallic nanoparticles. They are widely used for diverse biomedical applications owing to the simplicity of their synthesis, their biocompatible nature, the ease with which their surfaces can be modified, and their remarkable optical activities [18]. The tuning of the morphological and physical attributes of AuNPs can be used to easily adjust their optical activities from the visible to the NIR spectrum. In the NIR region, materials exhibit low absorption due to the more profound penetration abilities of NIR light compared to visible light. However, a combination of AuNPs with PTT has improved RA therapy via reducing the levels of inflammatory cytokines [16]. Li and colleagues developed hydrogels of hyaluronic, acid-modified, Triptolide-loaded AuNPs to treat RA in a CIA mice model [19].

Triptolide (TP) is a disease-modifying anti-rheumatic drug (DMARD). Owing to the NIR irradiation of AuNPs, TP can be released to specific inflammatory joint tissues and improve the treatment of RA in vivo. Moreover, these modified materials successfully minimised the invasion and migration of RA fibroblast-like synoviocytes in vitro. NIR irradiation increased the temperature and generated heat that accelerated the release rate of TP from the hybrid hydrogels (Figure 5). This investigation suggests that the TP/Au/HA hybrid hydrogel combined with PTT is more effective in terms of RA therapy than conventional treatment with TP alone. Moreover, their results also revealed minimal dose-related side effects. 

Another study demonstrated the combined effect of PTT and MTX-loaded PLGA Au/Fe/Au NPs conjugated with arginine-glycine-aspartic acid (RGD) for RA therapy applied to CIA mice [20]. This study also showed the enhanced release of the employed drug, namely, MTX, upon NIR irradiation, thus allowing the AuNPs to generate heat and offering sustained drug release at the inflammation sites of mice. In this case, Fe supports the NPs’ delivery to inflamed joints by applying an external magnetic field and enhancing their residence time in the joints. Moreover, this investigation suggests that the combination of targeted PTT and an external magnetic field improves MTX’s release and therapeutic effect in vivo (Table 1).

### 3.4. Polymer-Based Copper Nanomaterials 

Like AuNPs, Cu-based NPs also play a vital role in various biomedical applications because of their superior biocompatibility, ease of synthesis, and PTT alteration efficacy. In addition, Cu aids osteogenic differentiation and bone development and improves chondrocyte proliferation [21]. Therefore, Cu is a promising candidate for repairing the RA-induced breakage of bone and cartilage. 

The combination of PTT and photodynamic therapy (PDT) with Cu_7.2_S_4_ NPs under the influence of NIR reduced the intrusion of synovial liquid and enhanced anti-inflammatory effects in the arthritic joints of CIA rats. This also inhibited C-reactive protein and tumour necrosis factor (TNF-α). As the concentration of NPs increases, the solution temperature rises from 41 °C (125 mg/mL) to 55 °C (500 mg/mL). Singlet oxygen and reactive oxygen species generation were investigated to study their photodynamic effects.

Moreover, a comparative study between the impact of (NaCl + NIR) and (Cu_7.2_S_4_ NPs + NIR) on the inflamed paws of rats was reported. In this study, the researchers revealed the significantly greater therapeutic effect of (Cu_7.2_S_4_ NPs + NIR) compared to (NaCl + NIR). In another approach, Huang and associates designed a CuS shell and Au nanorod core (Au NR@CuS) together with PTT and PDT for RA treatment to reduce the hyperplasia of the synovium [9,22]. The localised surface plasmon resonance of both Au and CuS in the Au NR@CuS yolk-shell NPs was sufficient to enhance the efficacy of PTT, which may ultimately prove fruitful for the development of a cure for RA and the prevention of normal cells from being damaged.

Moreover, this significant core–shell structure was surface-modified with hyaluronic acid (HA) and vasoactive intestinal peptide (VIP) for the loading of MTX and its release in a controlled manner to a specific arthritic site. VIP-HA-Au NR@CuS-MTX demonstrated excellent therapeutic effects in vivo and higher NIR absorption than Au NR@CuS. Furthermore, both HA and VIP played a significant part in controlling or regulating the release of MTX. Hence, this investigation suggests that the synergistic activity of NPs and PTT improved RA treatment (Table 1).

**Table 1 pharmaceutics-15-00735-t001:** Polymer-based nanomaterials used in photothermal combined therapy of arthritis.

Photothermal Therapy	Objective of Investigation	Method of Synthesis	Drug Used	In Vitro Model Used	Animal Model Investigated	Outcome of Investigation	Ref
Polymer-based nanomaterials	Establish the controlled release of drug at specific arthritic sites using PCL-PEG micelles	Film dispersion	Dexamethasone	Murine macrophage Raw264.7 cell line and human umbilical vein endothelial cell	Rats with adjuvant-induced arthritis	Reduced joint swelling, bone erosion, and inflammatory cytokine expression in both joint tissue and serum	[12]
	Methotrexate-loaded PCL–PEG–PCL) triblock copolymer against RA	Copolymerization reaction and precipitation	Methotrexate	Fresh human blood and non-activated and lipopolysaccharide-activated macrophages	Mouse with RA	MTX-loaded nanomicelles proved to be a promising agent against RA	[13]
	To evaluate the influential role of PLGA NPs coated with anti- cyclooxygenase-2 (COX2) siRNA in arthritis treatment	Water-in-oil-in-water solvent evaporation technique	Dexamethasone	Human chondrocyte cell line (C28/I2)	-	Synergistic action of dexamethasone and COX-2 siRNA treatment reduced the expression of inflammatory and apoptosis-related factors produced in C28/I2 cells	[23]
	Evaluation of QRu-PLGA-RES-DS NPs) for effective arthritis therapy	Precipitation and sonication technique	Resveratrol	RAW 264.7 and HUVECs cells	CIA mice	QRu-PLGA-RES-DS NPs effectively treated RA after eliminating the inflammatory response	[24]
	NO-Hb@siRNA@PLGA-PEG (NHsPP) was utilised for osteoarthritis therapy	Minor medication, precipitation, and conjugation processes	Nitric oxide	RAW 264.7 cells and HEK 293T)	Mice	The therapeutic effect of the NHsPP NPs was significantly enhanced compared to the treatment groups using only NO, siRNA, or PTT.	[14]
Polymer-NIR780 bioconjugate	To investigate nanogold-core multifunctional dendrimer to establish photothermal therapy of RA	Citrate reduction method and conjugation process	Methotrexate	Mouse Macrophage RAW264.7 cells	-	Multifunctional targeted NPs proved to be potential therapeutics for the improved treatment of RA	[16]
Gold nanomaterial	Development of MTX-loaded poly(DL-lactic-co-glycolic acid) Au half-shell NPs (MTX-PLGA-Au) for arthritis treatment	Turkevich process for Au synthesis followed by MTX-PLGA conjugation	Methotrexate	Joint tissues were extracted for histological study	CIA Mice	This drug delivery system proved to be effective and minimised dosage-related MTX side effects in the treatment of RA	[25]
	Hyaluronate–gold nanoparticle/Tocilizumab (HA-AuNP/TCZ) complex was prepared for RA	Thiolated HA (HA-SH) was synthesised by reductive amination and conjugated with Au prepared using the Turkevich method	Tocilizumab	HUVECs cells	CIA Mice	Ha/Au/TCZ complex can be used for RA as well as other therapeutic applications	[15]
	Improvement of (MTX)-loaded PLGA) gold (Au)/iron (Fe)/gold (Au) half-shell nanoparticles conjugated with arginine–glycine–aspartic acid (RGD) for magnetic targeted chemo-photothermal treatment of RA	MTX-PLGA was prepared by a solvent evaporation method, and RGD was conjugated with MTX-PLGA Au/Fe/Au NPs	Methotrexate	-	CIA Mice	The combined effect of NIR irradiation and external magnetic field enhanced the therapeutic effects of NPs with an MTX dosage of only 0.05% dosage compared to free MTX therapy for the treatment of RA	[20]
	Hydrogels of hyaluronic acid hybridised with triptolide/gold nanoparticles for targeted delivery to rheumatoid arthritis-affected regions combined with photothermal therapy	Hyaluronic acid (HA) hydrogels loaded onto an RGD-attached gold nanoshell containing TP are prepared	Triptolide (TP)	-	CIA mice	Targeted photothermal-chemotherapy using hybrid hydrogels for the treatment of RA is an effective strategy that can maximize the therapeutic effects and reduce dose-related side effects.	[19]
Copper nanomaterial	To investigate combined effect of Au, CuS, and PTT on RA therapy	Seed-mediated and conjugated process	MTX	Murine fibroblast-like synovial cells (FLS) and mouse fibroblast (L929)	CIA rats	VIP-HA-Au NR@CuS-MTX revealed excellent therapeutic effects in vivo and higher NIR absorption than Au NR@CuS. Both HA and VIP played a significant part in controlling or regulating the release of MTX	[9]
Iron oxide nanomaterials	Synthesis of superparamagnetic IONPs to evaluate the efficacy of photothermal effect in arthritis treatment.	Solvothermal method	-	Cytotoxicity study (MTT assay) -RAW 264.7 cells	(CIA) mouse model	The prepared Fe_3_O_4_ nanoparticles showed promising results, with a size of 220 nm, high photothermal efficiency, and better targeting in the inflamed joint, thereby alleviating the symptoms of RA.	[26]
Black phosphorus nanosheets	Preparation of BPNs/Chitosan/PRP to study their synergistic effect with PTT and PDT therapy for improving osteogenesis in the treatment of RA.	Liquid exfoliation method	Methotrexate	Cytotoxicity assay using CCK-8 standard method (RAW264.7 cells, L929 cells and MSC cells)	(CIA) mouse model	Based on the distinguished concurrent PTT and PDT attributes of BPNs, chitosan/PRP thermos-responsive hydrogel effectively removed proliferating synoviocytes. In addition, BPNs accomplished calcium-extracted biomineralization via the in situ phosphorus-driven activation of BPNs in the target physiological microenvironment, which affected the entire course of treatment and provided improved therapeutic outcome.	[27]
Quantum dots	Synthesis of FAGM involved the following steps: 1. Formation of CTAB-coated gold nanorods (GNR) 2. GNR and methotrexate were coated with mesoporous silica shell 3. Synthesis of folic acid-functionalised GM.	1. Formation of CTAB-coated gold nanorods (GNR). The GNRs were prepared via the reduction of HAuCl_4_ with sodium borohydride. Afterward, CTAB, HauCl_4,_ AgNO_3_, and H_2_SO_4_ were added to the above solution in a certain amount. The solution was stirred at 28 °C for 30 min. Then, ascorbic acid was added dropwise, and a colour change was observed from light yellow to a state of colourlessness. The solution was kept overnight and stirred at 28 °C. 2. The prepared CTAB-GNR was again centrifuged and then resuspended. TEOS was added as a silica source and stirred for 30 min. Centrifuged and removed excess CTAB and FAGMs were collected. 3. Methotrexate was mixed with the prepared FAGM, and MTX-FAGM was formed.	Methotrexate	Cytotoxicity assay RAW 246.7 cells	Adjuvant-induced arthritis (AIA) rat model	In conclusion, for the therapy of RA, nanoscale MTX-FAGMs were prepared and their precisely targeted cytotoxicity towards active macrophages was confirmed under NIR laser irradiation. Synergistic action was observed between the photothermal therapy and chemotherapy	[28]

Abbreviations used in above table: PCl-PEG: (poly (ethylene glycol)-block-poly (ε-caprolactone); MTX: Methotrexate; PLGA: poly(DL-lactic-co-glycolic acid); NO: Nitric oxide, Hb: hemoglobin, SiRNA: Small interfering ribonucleic acid, PEG: polyethylene glycol; PTT: Photothermal therapy; CIA: Collagen-induced arthritic mice; FLS: Fibroblast-like synovial cells; CuS: Copper sulphate; TP: Triptolide; HA: Hyaluronic acid; RGD: Arginine–glycine–aspartic acid.

### 3.5. Iron Oxide Nanomaterial 

Due to their outstanding biocompatibility and low toxicity, iron oxide NPs (IONPs) have generated a great deal of interest with respect to biological applications. One intriguing characteristic of IONPs is their special superparamagnetism, which allows them to simultaneously function as heat therapy transducers, carriers for targeted drug delivery, and magnetic resonance imaging (MRI) contrast agents for diagnosis. Iron-based NPs can be used to magnetically transport NPs to a targeted region for the more concentrated, more potent, and longer-lasting retention of NPs in the targeted tissue and, subsequently, improved therapeutic efficacy [29]. IO has received FDA (Food and drug administration) approval as a contrast agent for MRI due to the superior biocompatibility of its iron-based composition [30]. IONPS are coated with different polymer-based coatings that help them stabilize. Examples include various polymers such as alginate, chitosan, dextran, poloxamers, polyvinylpyrrolidone (PVP), polyacrylic acid (PA), polyethylene glycol (PEG), and polyvinyl alcohol (PVA) [28]. As these polymers were found to be biocompatible in nature, they are widely used to stabilize IONPS.

Researchers found that colloidal gold-coated superparamagnetic IONPs, also known as AuSPIONs, had a significant therapeutic effect on RA murine models and also caused a reduction in the circulation of substantial organs [31].

To improve the targeting efficacy of SPIONs, researchers have studied the effects of NPs’ size on the efficiency of PTT for arthritis treatment, as shown in Figure 6 [26]. They have studied the size-based influence of Fe_3_O_4_ NPs in the range of 70–350 nm. However, it was observed that smaller NPs were quickly engulfed by the normal cells, leading to off-targeting. While the NPs with 220 nm size showed a rise in temperature under laser irradiation and gave exact targeting, they could not be engulfed by the normal cells. Still, they easily penetrated the inflamed arthritic tissue and were retained within it with as little difficulty. As a result, the use of NPs of the appropriate size benefits the PTT of RA, which should be appropriately tuned to produce the desired results.

### 3.6. Black Phosphorus Nanosheets 

Black phosphorous nanosheets (BPNs) are superior photo-responsive materials. They are widely used because of their excellent phototherapy effects, strong biocompatibility, and promising osteogenic qualities; they have drawn a great deal of attention [32,33]. During NIR irradiation, black phosphorous nanosheets can convert light energy into heat energy, thus providing a practical photothermal effect. In addition to being necessary for the human body, particularly in the bones, phosphorus is also a component of natural resources [34].

Recently, researchers prepared a black phosphorous nanosheet-based thermos-responsive hydrogel with the help of platelet-rich plasma (PRP)-chitosan as a temperature-responsive polymer [27]. It was found that BPNs can generate reactive oxygen species after irradiation with an NIR laser at 808 nm to remove hyperplastic synoviocytes. The thermos-responsive hydrogel could also regulate the release of the MTX by providing the essential elements for osteogenesis from the degraded products of the BPNs. Further, an in vivo study revealed a reduction in oedema in the developed CIA mouse model [17]. A few other recent studies are discussed in Table 2.

### 3.7. Use of Quantum Dots Together with Photothermal Therapy of Arthritis

Semiconducting polymeric quantum dots are a new class of functional nanomaterials being developed for photothermal therapy. Recently, scientists have prepared folate receptors that target semiconductive polymeric quantum dots [28]. To stabilize the quantum dots, they used cetrimonium bromide (CTAB) as a stabilizer. In addition to this stabilizer, the prepared quantum dots were further surface-functionalised to form a structure similar to mesoporous silica. This mesoporous silica nanoparticles acted as a reservoir for the hypoxia-activated prodrug Tirapazamine. These prepared polymeric quantum dots produced a localised photothermal effect in the activated macrophages and generated reactive oxygen species during irradiation with an NIR laser 808 nm, thus reducing the symptoms of arthritis.

## 4. Biological Applications of Polymer-Based Nanomaterials for PTT of Arthritis

### 4.1. Photothermal Therapy of Arthritis

PTT has recently received recognition as an effective RA treatment method (Gadeval A, 2021). Dual or multiple stimuli-responsive drug delivery systems, compared to single stimuli-responsive drug systems, offer greater control over medication release since they are more flexible and sensitive in terms of reacting to inflammatory situations. In one study, the investigators provided a method for constructing a dual or multiple-stimuli-responsive drug delivery system to treat RA. In the past, lipid NPs were produced by assembling triglycerol monostearate (TGMS) and 1,2-rdistearoyl-sn-glycero-3-phosphoethanolamine-carboxyl (DSPEPEG). In the current study, the matrix metalloproteinases’ cleavable TGMS segment was conjugated with PEG-COOH using 3-amino phenophenyl boronic acid (PBA) as a linker.

Compared to other stimuli-responsive drug delivery systems that have been previously discussed in the literature, the PEG has stealth features. They show that the PEG shell’s defence stabilizes Dex-loaded PPT micelles’ circulation. In addition, it accurately recognizes scavenger receptors through ligand–receptor interaction, wherein hydrophilic block Dextran Sulfate (DS) was selected as the active target. In one study, Polycaprolactone (PCL) and DS were combined to generate DS-b-PCL, which demonstrated improved biodistribution in the rear feet of CIA mice following each systemic injection compared to little in wild-type animals. Wang et al. [12] demonstrated that Dex-loaded PCL-PEG micelles significantly reduced bone resorption and joint swelling in arthritic rats at a low Dex dose without causing adverse effects.

In order to deliver MTX, researchers have conjugated a sialic acid-dextran-octadecanoic acid to MTX and prepared a MTX-loaded micelle. These micelles were able to enhance bone healing while also obtaining an anti-inflammatory benefit. Through the interaction of sialic acid and vascular endothelial cells, this MTX-loaded micelle greatly increased the drug’s accumulation in arthritic joints while showing good therapeutic advantages and few side effects. Effective therapies for RA include PTT and antioxidant combination therapy using Pd and Se NP co-delivery devices [37]. During the simple polymerization process, poly (ethyleneimine) (PEI) was used as a stabilizing agent that produced polypyrrole (PPy) NPs. The produced PPY-PEI NPs showed good photothermal performance and a sizable level of NIR absorption. For the treatment of RA, a strontium ranelate-loaded (SrR) methylcellulose hydrogel and PPY-PEI were used [38].

MTX and gold-based nanomaterials are routinely used for treating RA using PTT and chemotherapy, which induce a significant drop in inflammatory cytokine levels. When combined, gold nanorods and shells produce heat. Therefore, they are treated with NIR light and have improved therapeutic efficacy when combined with MTX. Magnetic iron oxide (Fe_3_O_4_) NPs have been utilised to treat and diagnose RA. Their biodistribution is ultimately impacted by the differences in their properties concerning size, charge, roughness, and elemental makeup, as depicted in Figure 7. Fe_3_O_4_ NPs exhibit remarkable biocompatibility and stability upon exposure to NIR light at higher temperatures [17].

Platinum NPs with a monodisperse octahedral crystal structure are uniformly layered on the outer surface of a metal-organic framework (MOF). They depend on Perovskite quantum dots and MOF-loaded polydopamine to enable H_2_-based thermal therapy. H_2_ can rapidly diffuse across cell membranes and acts as an antioxidant stress agent that reduces harmful reactive oxygen species and protect cells from damage. H_2_ is more effectively generated through photocatalysis. Additionally, H_2_ can protect healthy cells from the damage caused by overheating [38].

### 4.2. Chemophotothermal Therapy of Arthritis

Chemotherapy is the primary method of treatment for RA, along with surgery. Most patients can enter remission after employing treatments, biological agents, glucocorticoids, non-steroidal anti-inflammatory medications (NSAIDs), and disease-modifying anti-rheumatic pharmaceuticals. NSAIDs such as aspirin reduce the signs and symptoms of inflammation by inhibiting COX-2 activity [31]. The curative effects of DMARDs such as MTX are also influenced by immunosuppression. However, because of its slow therapeutic effect, DMARD treatment in clinics must be coupled with other drugs. Additionally, this treatment may cause digestive issues, bone marrow suppression, liver, and kidney damage.

One of the most well-supported nanocarriers, liposomes, have been used to successfully distribute Dex, an anti-RA drug, to RA patients. Liposomes incorporating synthetic materials of 100 nm in diameter and surface functionalised with 10% 5 kDa PEG that is slightly negatively charged have been shown to improve their in vivo half-life and capacity to target inflamed joints. In addition, the anti-arthritic activity of Dex is significantly boosted after being enclosed within liposomes according to pharmacodynamic studies using CIA mice [39]. For the treatment of RA, numerous studies have shown the synergistic advantages of combining PTT, chemotherapy, and targeted delivery. Therefore, the use of metallic NPs for specific treatment in combination with targeted drug loading and a hyperthermia agent is conceivable.

In this regard, Kim et al. [20] used PLGA loaded with gold [Au]/iron [Fe]/coupled RGD with MTX to maintain a high MTX concentration in painful joint regions. NIR radiation accelerates the release of MTX from PLGA NPs and induces localised heat generation due to Au half-shells’ NIR resonance. The Fe half-shell layer also enables the magnetic transport of the NPs to the target site via the use of a magnetic field, and their retention can be improved in the presence of an external magnetic field. These NPs were injected intravenously into CIA mice. NIR absorbance and MRI revealed an accumulation of NPs in the inflamed paws, and an external magnetic field increased this accumulation [40].

### 4.3. Image-Guided Photothermal Therapy of Arthritis

An innovative method among the phototherapeutic approaches is the image-guided application of light, in which only the target region with a high photosensitizer concentration should be illuminated. Several procedures have involved the provision of WS2 or Au NPs to CT-PTT [41]. Trigger-responsive engineered nanocarriers (ENCs), a recent advancement in the treatment of RA, assist in resolving issues typically associated with conventional medicine-delivery techniques. Due to image guidance, the development of specialised ENCs that improve drug targeting while reducing off-target toxicity is now possible. Numerous imaging methods have been utilised to examine inflammatory joints in RA patients using a range of probes, including computed tomography (CT), positron emission tomography, photoacoustic (PA) imaging, MRI, ultrasound, and X-ray.

A multipurpose imaging tool that can “picture” and “trigger” the targeted, controlled release of medication from ENCs is required. Only a handful of proof-of-concept studies have examined the efficacy of these RA therapeutic options, which are currently in the early stages of development.

Due to the progressive, debilitating nature of the disease; its unknown cause; and its initial similarities to other inflammatory diseases, RA is notoriously difficult to diagnose in its early stages. NIR-II photoacoustic molecular imaging (PMI) is being developed as a promising new strategy for the precise diagnosis and direction of RA treatment due to its exceptional sensitivity and specificity upon deep penetration.

In one study, TCZ, a targeted anti-rheumatic drug, was combined with polymer NPs to provide the first NIR-II-based theragnostic nanoplatform, or TCZ-Polymer NPs, for the PA-imaging-guided therapy of RA. NIR-II PMI data have shown that TCZ-Polymer NPs have better targeting capabilities for the effective non-invasive diagnosis of RA joint tissue, with a high signal-to-noise ratio (SNR) of 35.8 dB in 3D PA tomography images. Additionally, it has been demonstrated that the results of clinical micro-CT and histological analyses are in agreement with respect to the NIR-II PMI treatment evaluation of RA mice.

Six Harlan rats developed arthritis in their knee and ankle joints with the assistance of peptidoglycan polysaccharide polymers. Three rats comprised the untreated controls. Before and up to 24 h after the intravenous administration of 10 mg/kg ICG, optical imaging of the knee and ankle joints was performed using an integrated OI/X-ray system. The fluorescence signal intensities of the joints with and without arthritis were analysed for any discernible differences using generalised estimating equation models. Joint samples from the knee and ankle were further examined and evaluated using histology (Meier, 2010).

Nimesulide and low-dose MTX would be present in therapeutic RGD-mediated polymeric micelles that would be used to actively and passively target RA. As a result, they adhered RGD to NHS-PEG3400-PLA2000, an FDA-approved amphiphilic copolymer that is biocompatible and biodegradable. Additionally, we conducted real-time fluorescence imaging analysis to examine the in vivo distribution of the fluorescence-labeled R-M/N-PMs and in vivo investigations in a rat model of adjuvant-induced arthritis to evaluate the anti-inflammatory activity of R-M/N-PMs.

This problem may be resolved through the creation and application of magnetic NPs, namely, ultrasmall paramagnetic iron oxide (USPIO) (Over the past ten years, many researchers have carefully investigated the application of USPIO to MRI. Highly magnetically sensitive hydrophobic NPs are destroyed by high temperatures. However, the hydrophobicity of the NPs necessitates further modification before they can be transferred to the aqueous phase, which makes the NMs extremely unstable or agglomerated. By using in situ polyol polymerization, an amphiphilic block polymer called polyethylene glycol-tert-butyl polyacrylate (PEG-b-PAA) was created, coated with USPIO, and conjugated with folic acid (FA) with a targeted function towards the PEG-b-PAA/USPIO surface; subsequently, researchers were able to develop a highly stable and focused nanocomposite contrast agent (referred as FA-PEG-b-PA). Herein, the authors also investigated the impact of FA-PEG-b-PAA/USPIO on RA’s early diagnosis.

When PTT is used with external magnetic field-targeted therapy, therapeutic efficacy increases, and adverse effects are reduced with fewer doses. Additionally, iron-based parts enabled T2-weighted magnetic resonance imaging (MRI) in vivo, thus supplying the RA treatment’s guiding and monitoring capability. Magnetic iron oxide NPs (IONPs) have a unique ability to convert light to heat. The FDA has approved IO as a contrast agent for MRI because of the high biocompatibility of its iron-based composition. Colloidal gold-coated superparamagnetic IONPs, also known as AuSPIONs, were found to have a significant therapeutic effect on RA murine models and provoke a reduction in circulation in the principal organs by [26] investigated the influence of 70–350 nm-sized Fe_3_O_4_ NPs on the effectiveness of PTT for RA treatment to improve biosafety and its targeting impact.

### 4.4. PDT Combined Photothermal Therapy of Arthritis

Photodynamic therapy, which uses photosensitizers or non-toxic dyes along with safe, visible light, has been used for more than a century, but is just now gaining popularity [42,43]. In addition to oxygen and light, photosensitizers are among the three essential components of PDT. According to their definition, these dyes are compounds that may absorb light of a specific wavelength and cause photochemical or photophysical processes. The three non-toxic components that make up the molecular basis of photodynamic therapy—a suitable light wavelength, a photosensitizer, and oxygen dissolved in cells—interact to generate the intended effects only when they are present in pathogenic tissues [44].

PDT primarily relies on the presence of oxygen present inside the cells. When a photosensitizer enters the cell, it is exposed to light at a wavelength that coincides with the photosensitizer’s absorption spectrum. The absorption of a photon causes the excitation of the photosensitizer from the ground state to the excited state. The energy is released as fluorescence while the residual energy causes the excitation of the photosensitizer towards the excited triplet state, thus generating phosphorescence [45]. Co-delivery for trigger response is the most frequently used technique, particularly when combined with PTT. There are two light-triggered therapies: PTT and PDT. To properly treat RA, a combination of PTT and PDT may provide more significant bone and cartilage preservation, less synovial invasion, and a more substantial anti-inflammatory effect (Figure 8). Important light triggers for both treatments include photothermal agents and photosensitizers [35].

As a result, they are frequently co-delivered with conventional agents. Palladium (Pd) blue was the term given to the hexagonal palladium nanosheets developed by Chen et al. [46] with a blue colour. The ultrathin Pd nanosheets display precise and predictable NIR surface plasmon resonance peaks and significant photothermal effects in vitro. They can be utilised as photothermal agents in PTT therapy. However, it is challenging to use inorganic materials alone due to their poor biocompatibility. Considering the photothermal properties of Pd, Xu et al. [21] developed and manufactured nanosheets to treat RA by targeting inflammatory cells and controlling MTX release with Pd particles and RGD peptides. RGD peptides can change Pd NPs, an inorganic photothermal agent, such that their capacity for targeting is enhanced and the disadvantages of inorganic photothermal agents’ poor biocompatibility are overcome. The nanosheets showed exceptional PTT efficacy in terms of treating RA-afflicted animals, as it accumulated and significantly reduced inflammatory responses, cartilage degradation, and MTX toxicity in inflamed paws. This indicates that the nanosheets have excellent in vitro anti-inflammatory properties, superior stability, and photothermal conversion efficiency.

Chen et al. [46] reported the design and synthesis of PdNps (Pd-Cys@MTX@RGD) as a nanotherapy agent that can target inflammatory cells and control MTX release. The Pd-Cys@MTX@RGD nanosheets served as an NIR photothermal agent with selective targeting potential towards peptides overexpressed on the surface of inflamed arthritic tissues. Pd-Cys@MTX@RGD nanosheets were observed to mediate the controlled release of MTX upon NIR 808 nm laser irradiation with a power density of 0.3 W cm^2^. This was associated with a significant reduction in the toxicity issues associated with MTX. It was advocated that, like MTX, other therapeutic agents can be combined with PTT to amplify its medicinal benefits (Figure 9). The agent in question inhibited the inflammatory response induced by vascular endothelial growth factor (VEGF) and IL-1β. This was correlated with its in vivo performance, which significantly inhibited RA and led to a reduced level of pro-inflammatory cytokines (TNF-α, COX-2).

PTT and antioxidant combination therapies employing Pd and Se NPs delivery devices are effective treatments for RA. The straightforward polymerization procedure that created polypyrrole (PPy) NPs employed PEI as a stabilizer. The created PPY-PEI NPs showed significant NIR absorption and exhibited good photothermal stability and performance [47]. A methylcellulose hydrogel co-loaded with PPY-PEI and strontium ranelate (SrR) was utilised to treat RA.

A unique method for the treatment of RA involved the use of CuS NPs based on copper NPs and a novel l-cysteine-assisted synthesis. In addition to stimulating chondrogenesis and osteogenesis, copper acts as a photothermal and photosensitizer. In vitro investigations have shown outstanding biocompatibility and excellent photothermal and photodynamic effects.

### 4.5. Gene Therapy-Combined Photothermal Therapy of Arthritis

RNA interference (RNAi) is a cellular process that silences messenger RNA (mRNA)-based genes post-transcriptionally, and small interference RNA (siRNA)-based silencing is only temporary. Hence, new methods have been devised and described to produce longer-lasting silencing. The use of short hairpin RNA (shRNA), for example, is a vector-encoded technique that can be employed for long-lasting, reliable cell silencing. Small interfering RNA (siRNA), which is highly specific and effective in terms of silencing genes, is a critical component that is mediated by RNA interference [28]. An intravenous dose of Cy5-labeled siRNA/WS was used by Pan et al. [27] to test the therapeutic activity of their wrap some (WS)-containing siRNA (with a core constituted of a cationic lipid bilayer and an siRNA complex enveloped in a neutral lipid bilayer in PEG over the surface). In addition, they used the arthritis scores among CIA mice to demonstrate the effectiveness of siRNA’s ability to target tumour necrosis factor (TNF-/WS).

RGD-functionalised, siRNA-loaded poly (lactide-co-glycolide) NPs were created by Scheinman et al. [48] as a nanosystem for STAT1 siRNA delivery to the joint tissues of a CIA mice model. After siRNA was enclosed, the researchers examined the stability and nanoparticle-related properties. According to morphological analysis, RGD-NPs have a size range of 100–200 nm and a net positive charge, which may result from RGD functionalization. The presence of the RGD peptide on the NPs outer surface improves tissue uptake by 10–200 times in arthritis-prone mice, and PLGA NPs shield siRNA from serum breakdown. Compared to the control group, the group of arthritic mice that was treated with RGD functionalization showed improved lung delivery of NPs. For the treatment of arthritis, RGD-functionalised PLGA NPs encapsulating STAT1-targeted siRNAs may be more efficient, presumably by selectively inhibiting macrophage and dendritic cell activation.

For the treatment of RA, Park et al. created PLGA NPs that were loaded with Dex and siRNA. In this study, PLGA NPs were first loaded with Dex before being combined with PEI/siRNA [23]. Then, green fluorescence protein siRNA (GFP siRNA) and medications were transfected into the human chondrocyte cell line (C28/I2) to evaluate the co-delivery of siRNA and Dex. For the treatment of persistent skin inflammations, Desai et al. [49] created cyclic cationic head lipid–polymer hybrid nanocarriers (CyLiPns) packed with the analgesic Capsaicin (Cap) and anti-TNF siRNA (siTNF). A few of the photothermal combined therapies are illustrated in Table 2.

## 5. Conclusions

Polymer-based NMs and PTT have led to tremendous growth in nanomedicine, which has been proven through various publications and clinical trials over the past years. The combined effect of PTT and polymer NMs has revealed multiple advantages compared to conventional therapies such as radiotherapy, chemotherapy, and immunotherapy for treating diseases such as cancers and RA. Polymer-based NMs are cost-effective, biocompatible, and beneficial materials that are easily functionalised with functional groups. They can convert light to heat energy and thus enhance their own photothermal conversion efficiency. They are used as nanocarriers for controlled drug release that improve the efficacy of RA treatment and reduce the disease’s side effects. With the help of NMs, drugs bind to specific target sites, produce signals for imaging determinations, and generate ROS to support PTT and PDT performance. This review expounds on various polymer-based NMs for the photothermal therapy of RA, including Au, Cu, iron oxide, black phosphorous, quantum dots, and palladium. These NMs have achieved excellent improvement in the treatment of RA via their combination with PTT. However, many issues still need to be discussed, such as their low photothermal conversion efficacy, safety, and low in vivo efficiency, which limit their clinical applications. In addition, their accumulation in non-targeted cells or tissues is another problematic issue. Thus, besides their effective treatment efficiency, the interaction of such polymer NMs with tissues is a matter of concern that requires further investigation to determine their potential applications. In the future, we believe that nanotechnology and advanced polymer-based NMs will play a vital role in treating RA.

## Figures and Tables

**Figure 1 pharmaceutics-15-00735-f001:**
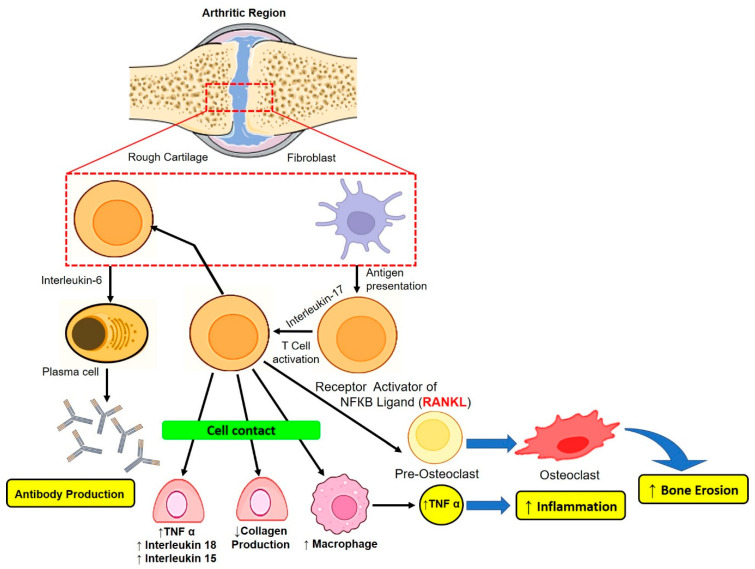
Pathophysiology of Arthritis.

**Figure 2 pharmaceutics-15-00735-f002:**
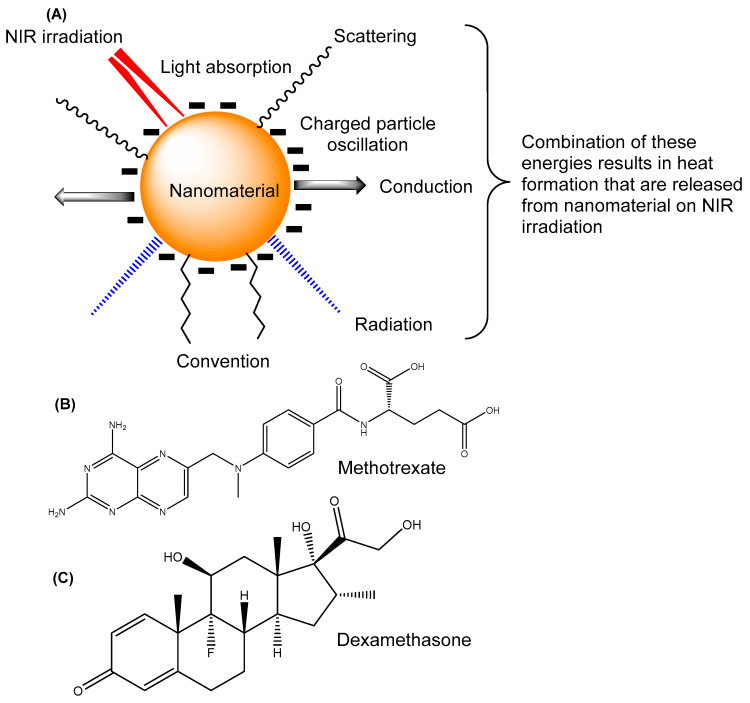
(**A**) Photothermal conversion of nanomaterial. In this process, heat is released in the form of radiation, scattering, charged particle oscillation, conduction, and convection, which are used for RA therapy and cause cell apoptosis; structures of (**B**) Methotrexate and (**C**) Dexamethasone.

**Figure 3 pharmaceutics-15-00735-f003:**
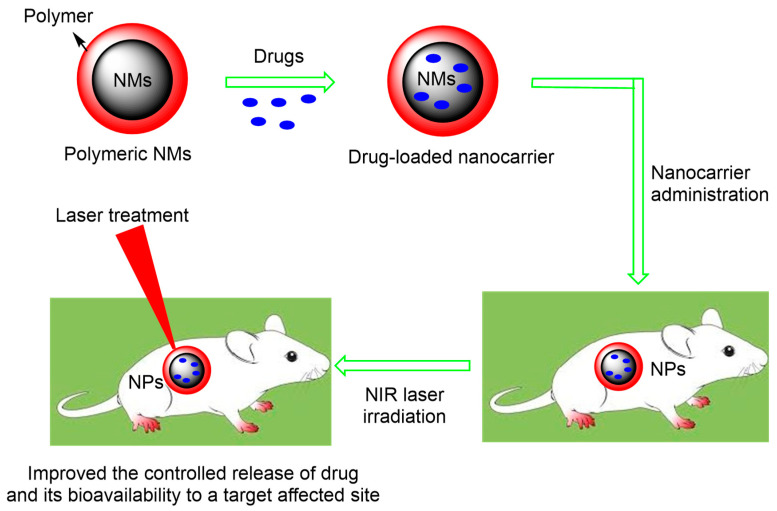
Scheme representing the use of polymeric nanomaterials as photothermal agents for arthritis therapy. Such polymeric nanomaterials enhanced DMARD’s bioavailability and sustained release and proved to be an effective RA therapy.

**Figure 4 pharmaceutics-15-00735-f004:**
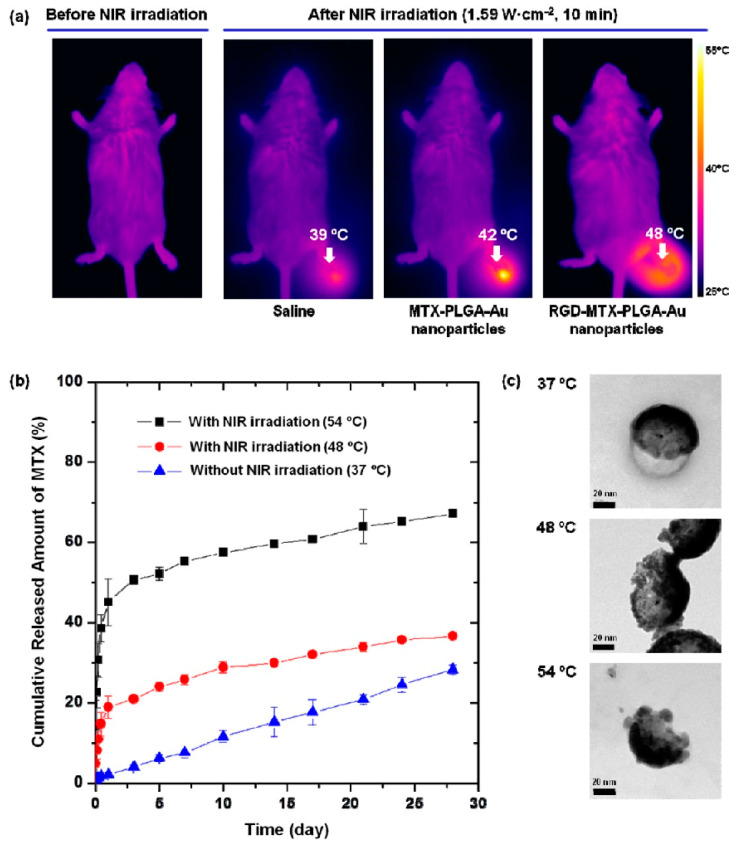
(**a**) Thermal images of CIA mice treated with saline, MTX-PLGA-Au nanoparticles (150 μL, 1 mg/mL dispersed in PBS), or RGD-MTX-PLGA-Au nanoparticles (150 μL, 1 mg/mL dispersed in PBS) before and after NIR exposure (1.59 W/cm^2^, 10 min) of the right paw. (**b**) Profiles of MTX release from RGD-MTX-PLGA-Au nanoparticles with and without NIR irradiation of 0.38 (48 °C) or 0.53 (54 °C) W/cm^2^ for 10 min at the initial time. Data represent mean values for n = 3, and the error bars represent the standard deviation of the means. (**c**) TEM images of RGD-MTX-PLGA-Au nanoparticles measured after MTX release experiments without (top) or with NIR irradiation of 0.38 (48 C, middle) or 0.53 (54 C, bottom) W/cm^2^ for 10 min at the initial time. Adapted with permission from reference [15]. Copyright 2022 ACS publisher.

**Figure 5 pharmaceutics-15-00735-f005:**
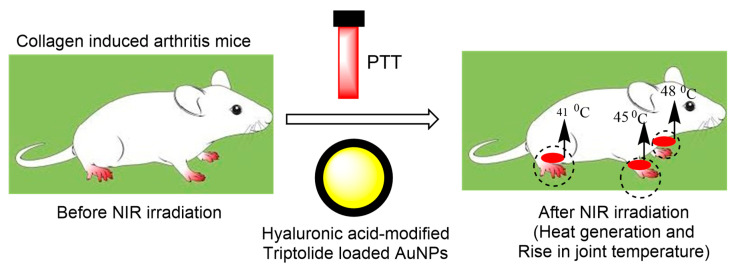
Indicates the synergistic activity of PTT and hyaluronic-acid-modified, Triptolide-loaded AuNPs for the treatment of CIA mice. The combined effect of PTT and drug-loaded polymer AuNPs has improved arthritis treatment via heat generation, which causes a temperature increment.

**Figure 6 pharmaceutics-15-00735-f006:**
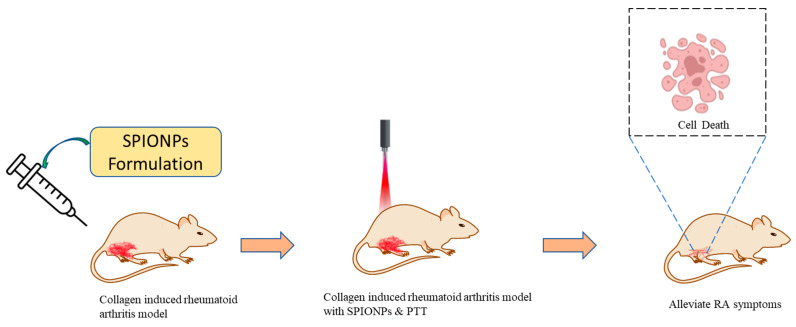
Injection of superparamagnetic IONPs (SPIONPS) combined with photothermal therapy in the collagen-induced arthritic model for the alleviation of arthritis symptoms.

**Figure 7 pharmaceutics-15-00735-f007:**
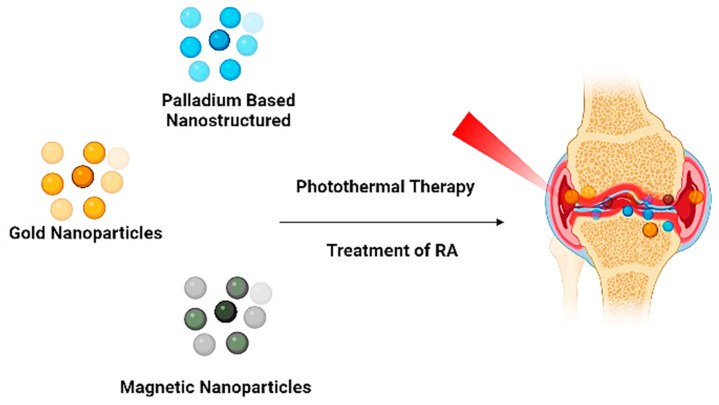
Photothermal Therapy.

**Figure 8 pharmaceutics-15-00735-f008:**
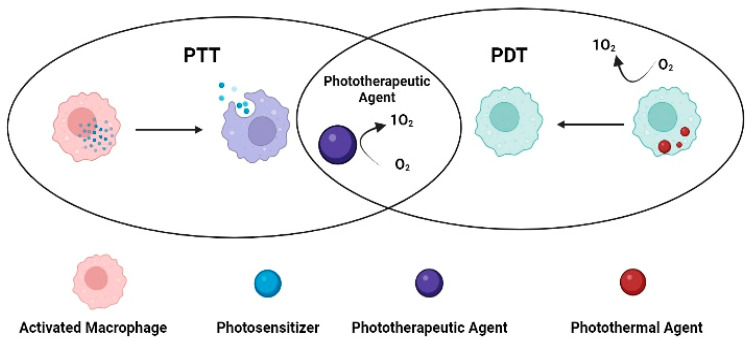
PDT combined photothermal therapy.

**Figure 9 pharmaceutics-15-00735-f009:**
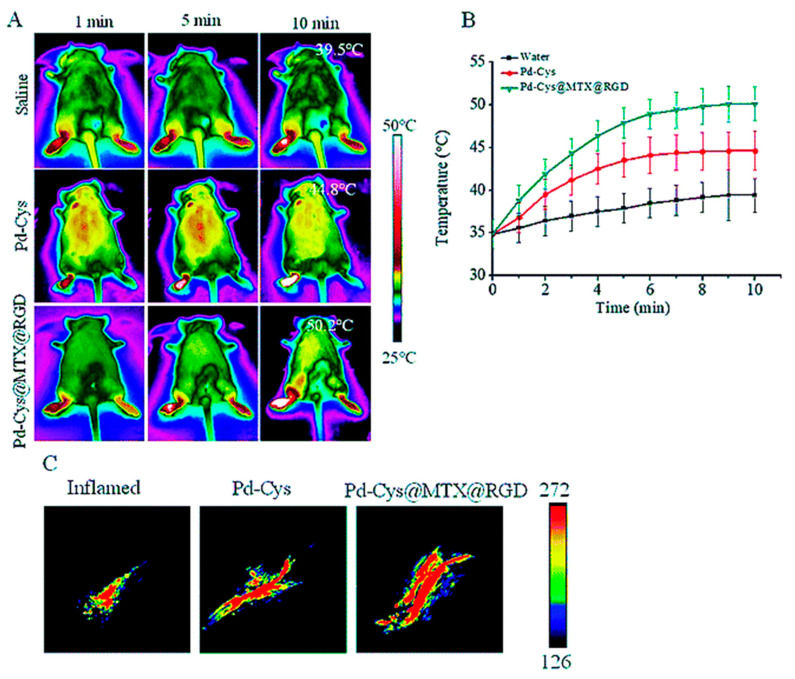
(**A**) Photothermal images of the Pd-Cys and Pd-Cys@MTX@RGD nanosheets in RA mice irradiated at 808 nm for 10 min (0.3 W cm^−2^). (**B**) Temperature profiles of the Pd-Cys and Pd-Cys@MTX@RGD nanosheets in RA mice irradiated at 808 nm for 10 min. (**C**) Photoacoustic signal images of the Pd-Cys and Pd-Cys@MTX@RGD nanosheets in RA mice excited with 808 nm irradiation. Adapted with permission from reference [46]. Copyright 2022.

**Table 2 pharmaceutics-15-00735-t002:** Polymer-hybridised nanomaterials in photothermal combined therapy of arthritis.

Photothermal Therapy	Nanomaterial Used	Combination Therapy	In Vitro Model Used	Animal Model Investigated	Outcome of Investigation	Ref
Photothermal Therapy alone	Palladium nanosheets		-	CIA Mouse	Pd-MTX nanosheets reduced the toxicity of MTX and prevented VEGF from proliferating as well as the generation of pro-inflammatory cytokines such as TNF-a and COX-2.	[17]
	Gold nanorods	PTT combined with chemotherapy	-	CIA mouse model	Aggregation of NPs increased in the presence of an external magnetic field, which raised the temperature in the exposed region and expedited the release of MTX.	[17]
	Magnetic iron oxide (Fe_3_O_4_) NPs	PTT combined with chemotherapy		CIA mouse model	Since NPs’ small size allowed them to enter inflammatory cells, more of them accumulated in inflamed joints.	[17]
Chemo-photothermal therapy	Gold half-shelled Nanoparticles	Chemotherapy with photothermal therapy		CIA mouse model	In the inflammatory joints, nanoparticle accumulation was elevated.	[15]
PDT combined Photothermal therapy	Palladium nanosheets	PTT combined with photothermal therapy	Fibroblast cells	CIA Mouse	Pd-MTX nanosheets reduced the toxicity of MTX and prevented VEGF from proliferating as well as the generation of pro-inflammatory cytokines such as TNF-a and COX-2.	[14]
	Copper-based Nanoparticles		-	CIA Mouse	Cu_7.2_S_4_ NPs act on joints that showed good bone retention and resembled normal joints	[35]
Gene therapy combined Photothermal therapy	Poly (lactide-co-glycolide) nanoparticles (NPs)	Gene therapy combined with photothermal therapy	-	CIA Mice	RGD functionalised PLGA nanoparticles encapsulating STAT1-targeted siRNAs may be more efficient, presumably by selectively inhibiting macrophage and dendritic cell activation.	[36]
	Dexamethasone-loaded PLGA nanospheres	Gene therapy combined with photothermal therapy	-	CIA Mouse Human TNF Transgene model	The effectiveness of PLGA-PEG NPs for the delivery of a therapeutic medicine in the affected tissues in rheumatoid arthritis was low, perhaps because of the leaky vasculature, angiogenesis, and associated ELVIS effect that occur in the affected joints.	[36]

## Data Availability

Not applicable.

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
