# Peer review of "Recent Advances in Polymer-Based Nanomaterials for Non-Invasive Photothermal Therapy of Arthritis"

_pharmaceutics, 2023, doi:10.3390/pharmaceutics15030735_

Round 1
Reviewer 1 Report
This work is interesting and worthy of publication. However, as far as the English language is concerned, it needs to be double-checked and verified by the authors.
The textual content of Figures 1 and 3 is not very clear, so please improve the quality of the images.
If images or data from other papers are cited, please clearly list the copyright notice.
Author Response
The authors response file is attached.

Reviewer 2 Report
In this review, the authors summarized recent advances in polymer-based nanomaterials for photothermal therapy of arthritis. They first gave a summary of the mechanism of photothermal therapy, and then different types of polymer nanomaterials and their biological applications were discussed. However, several issues need to be addressed:
1. Line 67: the term “NIR” needs to be defined first, as it may refer to different wavelengths in different contexts.
2. Section 2: the description of the physical process of PTT is not accurate. For example, “radiation” usually means emission of photons in photochemistry, which is not related to heat generation.
3. Section 3: in this section, it would be helpful if the authors can compare PTT properties (such as conversion efficiency, temperature achieved, and on/off properties) of different materials.
4. Section 3.1: the concept of “polymer-based nanomaterials” is not well-defined in this section. From the context, it seems that the term refers to small molecule encapsulated with a polymer. If so, some examples in section 3.2 (for example, MTX-loaded PLGA nanoparticles) should be moved to this section. To make it clearer, it’s helpful to define the concept first, or include a schematic representation of the structure.
5. Line 117: The sentence needs to be revised to make it clear. It’s confusing that the “NIR light” is a “sound response system”.
6. Section 3.2: the concept of “bioconjugate” is not defined here. Is the dye covalently linked to the polymer, or loaded in the polymer with non-covalent interactions? If both sections 3.1 and 3.2 discussed the same type of nanomaterials, these two sections can be combined.
7. Line 169: this sentence needs to be revised to make it clear.
8. Section 3.5: the only example in this section is not a polymer-based material. The nanomaterial is stabilized by cysteine, which is not a polymer. This section should be removed.
9. Examples from sections 3.6, 3.7, and 3.8 should also be included in table 1.
10. In section 3.6, the authors should emphasize which polymer is used to stabilize iron oxide nanoparticles.
11. Section 3.8: CTAB and mesoporous silica are not polymers. Although one might argue that mesoporous silica is a type of inorganic polymer, it seems that the focus of this review should be organic polymers. There are a lot of examples of polymer-stabilized quantum dots, and the authors should use those references as examples here.
12. Line 291: the introduction to PTT has been done in the second section, and this sentence is redundant.
13. Section 4: this section needs to be reorganized. It seems that it’s rare to find an example of PTT alone. If that’s the case, the authors should mention that PTT is always combined with other therapeutic methods, and then each section can focus on one type of therapy.
14. It might be helpful to give the structure of some small molecules mentioned in the manuscript, such as TCZ, MTX and Dex. It can be interesting to researchers with a chemistry background.
15. Section 4.4: the mechanism of PDT should be introduced in this section.
Author Response
The authors response file is attached.

Reviewer 3 Report
The manuscript intends to revise Polymer-Based Nanomaterials for Non-Invasive Photothermal Therapy of Arthritis. There is a misleading description in the abstract and within the manuscript saying that "polymer-based nanomaterials ... act as photothermal reagents having high absorption in the near-infrared region that can transform near-infrared light ..." This is not true.
The manuscript compiles only 31 papers which is quite short for a review. Besides it is prepared as a description of the papers without any discussion or comments on the advances achieved.
It is not adequate for publication in Pharmaceutics.
Author Response
The authors response file is attached.

Round 2
Reviewer 2 Report
The manuscript has been largely strengthened after the revision, and most of the issues have been resolved. Therefore, it should be accepted for final editing.
Reviewer 3 Report
the authors properly addressed the raised issues